# Intermediate Model Design in the Progressive Stamping Process of a Truss Core Lightweight Panel

**Zhilei Tian** [1] , **Chenghai Kong** [2] , **Wei Zhao** [3], **Jingchao Guan** [1] **and Xilu Zhao** [1,*]

1   Department of Mechanical Engineering, Saitama Institute of Technology, Saitama 369-0293, Japan; tzl880108@gmail.com (Z.T.); guanjingchao123@gmail.com (J.G.)
2   Topy Industries Co., Ltd., Tokyo 441-8510, Japan; tei-kou@topy.co.jp
3   Weichai Global Axis Technology Co., Ltd., Tokyo 107-0062, Japan; wdgazhaowei@weichaigbaxis.com
*   Correspondence: zhaoxilu@sit.ac.jp

**Abstract:** The truss core panel has been verified to be effective for structural weight reduction in former research studies. However, it is difficult to manufacture using the sheet metal pressing method because the forming height of the truss core panel is limited by the physical properties of the material. Although progressive stamping has been used to solve this problem, it is still difficult to practically use the truss core panel. In this study, the author proposed a manufacturing method and a hexagonal frustum intermediate structure to improve the forming quality of truss core panels using a progressive stamping method and verified its effectiveness through numerical analysis and prototype experiments. Compared to the conventional hemispherical intermediate model, the manufacturing process of the truss core panel using the proposed method was significantly improved.

**Keywords:** truss core panel; progressive stamping; intermediate model; lightweight structure; forming experimental research; forming numerical analysis

## 1. Introduction

Lightweight design has always been an important target in automobiles, aircraft, and many other manufacturing industries, and to achieve this target, various lightweight structures have been proposed and studied in the past several years [1–5].

The honeycomb core panel, which is a lightweight and rigid structural panel, is relatively easy to produce with various core sizes and panel dimensions [6,7], and is widely used in industries because of its excellent sound insulation property and bending rigidity [8–12]. However, because the regular hexagonal cross-sectional core and the surface plate are bonded by adhesive, it is difficult to apply the structure to an environment where the temperature changes rapidly, vibration is high, or shear load is applied [13–15]. Therefore, various sandwich core structures with higher strength and bending rigidity than honeycombs have been proposed [16–18]. Among them, the octet truss core structure has been extensively studied for its high strength and rigidity [19–21]. In addition, a lightweight truss core structure consisting of truss core panels has been proposed based on origami engineering [22], the static bending stiffness and collision energy absorption performance of truss core panels have been investigated, and the mechanical advantage of truss core panels as lightweight structures has also been proven [23–26] by simulation.

However, it is difficult to manufacture truss core panels with high quality and low-cost owing to their complex three-dimensional shape. Because of the characteristics of thin-walled panels made of thin steel plates, it seems appropriate to use the sheet metal press method to fabricate truss core panels; however, the convex part of the truss core panel must be formed as high as possible using the deep-drawing method to increase the bending rigidity [27,28]. However, by forming the truss core as much as possible, the problem of extreme thinning of the plate can be observed locally. Because there is a limit to the height at which thin steel plates can be deep drawn [29,30], a multi-step deep-drawing forming

method has been devised, and the results of a study on processing truss core panels using a progressive press-forming method with a hemispherical intermediate model have been published [31]. However, a stable press-forming method for truss core panels has not yet reached a practical level.

In this study, the author handles the problem of intermediate model design, which is the most important issue when processing truss core panels using the progressive press-forming method and proposes a design method that replaces the hexagonal pyramidal intermediate model with the hemispherical intermediate model. Prototype experiments and numerical analyses were conducted to verify the formation characteristics of the proposed hexagonal pyramidal intermediate model. In comparison with the conventional hemispherical intermediate model, it has been verified that the problem of extreme thinning in the local thickness of truss core panels can be improved by using the proposed hexagonal pyramidal intermediate model in progressive press forming.

## 2. Materials and Methods

### 2.1. Truss Core Panel

A truss core panel comprises convex sections, called truss cores, arranged in sequence on a flat plate, as shown in Figure 1. The truss core is a hexagonal pyramidal stereoscopic structure, whose side faces comprise three trapezoids and three rectangles. When two truss core panels are joined with opposite sides facing each other, a double-truss core panel is formed, as shown in Figure 1. When a load is applied to a truss core panel in the out-of-plane direction, it can be easily diffused to the entire truss core panel through the diagonal planes, where the truss cores support each other. It has been confirmed that double-truss core panels not only have good bending stiffness and vibration characteristics [32–34] but also excellent crash energy absorption performance in the in-plane direction [35,36].

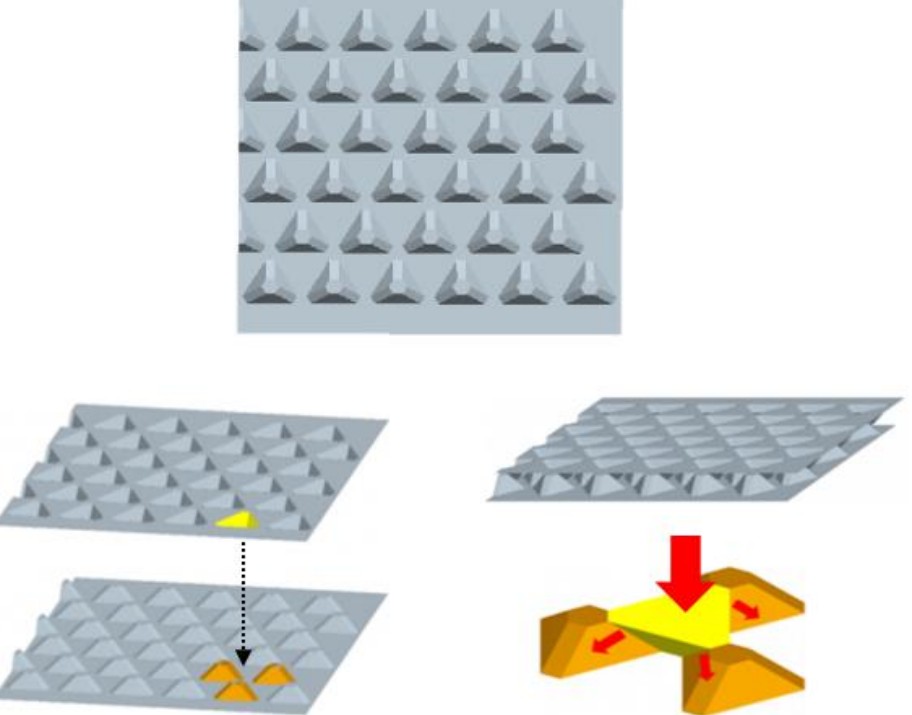

**Figure 1.** Truss core panel can diffuse the stress through diagonal surface of the truss core.

To increase the rigidity of the truss core panel, it is advantageous to increase the height of the truss core. However, when processing truss core panels using the deep-drawing method, there is a limit to the height at which the metal sheet can be deep drawn and formed.

### 2.2. Shape Parameters of Truss Core Panel

As shown in Figure 2, by adjusting only the rectangular contact surface and the hexagonal bottom and top surfaces, a truss core with a three-dimensional shape can be determined.

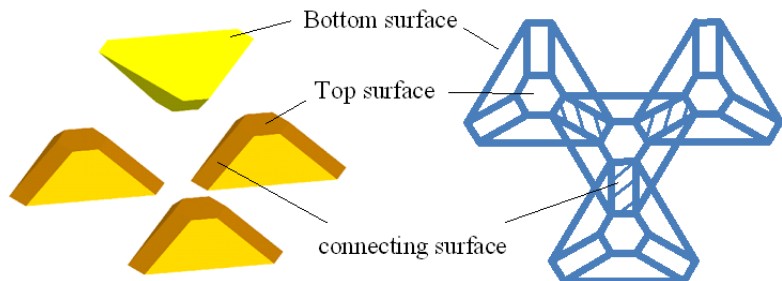

**Figure 2.** Basic shape of truss core panel.

A truss core panel was designed according to the procedure shown in Figure 3. In step 1, the plane is divided by an equilateral triangle with side length c. In step 2, the equilateral triangle is reduced or expanded from side length c to a. In addition, each acute angle was cut with width b in step 3, resulting in a hexagonal shape of the top and bottom faces, as shown in Figure 3. Finally, in step 4, only the top surface is extended vertically in the plane to height h to obtain a three-dimensional truss core shape, and width b is connected to obtain a rectangular contact surface.

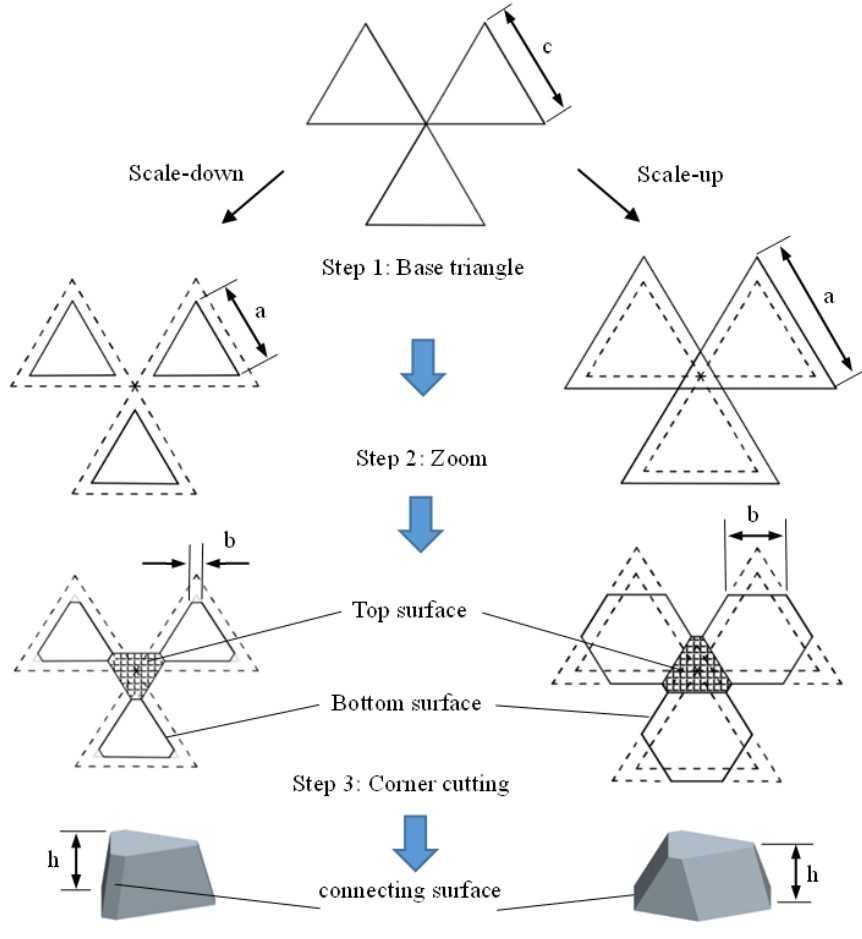

**Figure 3.** Design procedure of truss core.

The basic design parameters of the truss core shape are a, b, c, and h, which can be combined to form various truss core shapes, as shown in Figure 4. According to Figure 4, if the vertical axis $b/a = 0.5$ and the horizontal axis $a/c = 2$, the truss core will be a hexagonal prism. For a corner cut width of $b = 0$, the truss core panel will be a trigonal pyramid, and when $a/c = 0.5$ and $a/c = 1$, the truss core panel becomes a triangular prism and a triangular pyramid, respectively. In this study, the general truss core shape was targeted as $a = 35$ mm, $b = 5$ mm, $c = 35$ mm and h $= 9$ mm, as shown in Figure 5.

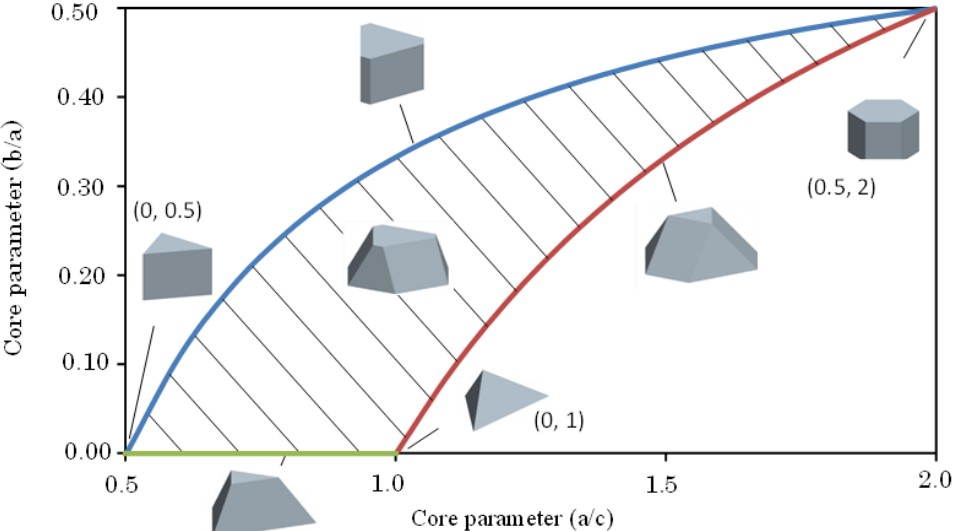

**Figure 4.** Relationship of core types and core parameters.

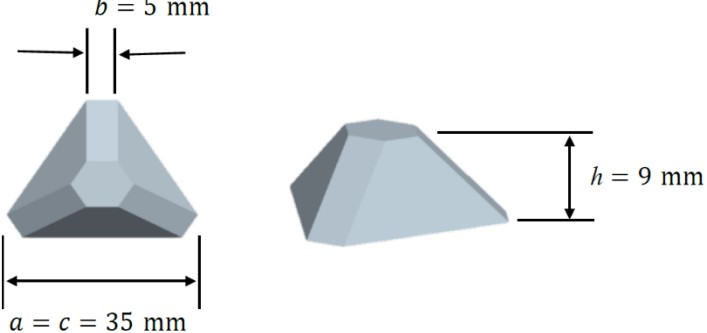

**Figure 5.** Truss core shape and parameters.

### 2.3. Progressive Press-Forming Method and Intermediate Model

To increase the height of the truss core panel, a two-step progressive press-forming method with the intermediate model shown in Figure 6 was proposed. The two-step progressive press-forming method involves two forming processes. In the first forming process, an intermediate model shape was formed from a flat plate, and in the second forming process, a truss core shape was formed from the intermediate model.

The procedure for processing a truss core panel using the progressive press-forming method is illustrated in Figure 6. The mold used for press forming consisted of three parts: punches and dies from left to right.

First, as shown in step (a) of Figure 6, press the punch downward against the plate set at position ① to form the plate into an intermediate model shape using the preformed punch and die ①.

Next, as shown in step (b) of Figure 6, the pre-formed plate is moved forward in one step. The positioning punch ② was fixed to the base plate via a spring. When the punch was pushed downward, positioning punch ② first contacted the plate. Because punch and die ② have the same shape as the intermediate model before molding, the elastic force of

the spring allows the plate to be set at the exact molding position. When the upper die is pushed downward, the elastic spring of punch ② is compressed and deformed, and simultaneously, the next intermediate model shape can be formed on the plate at position ① by punch and die ①.

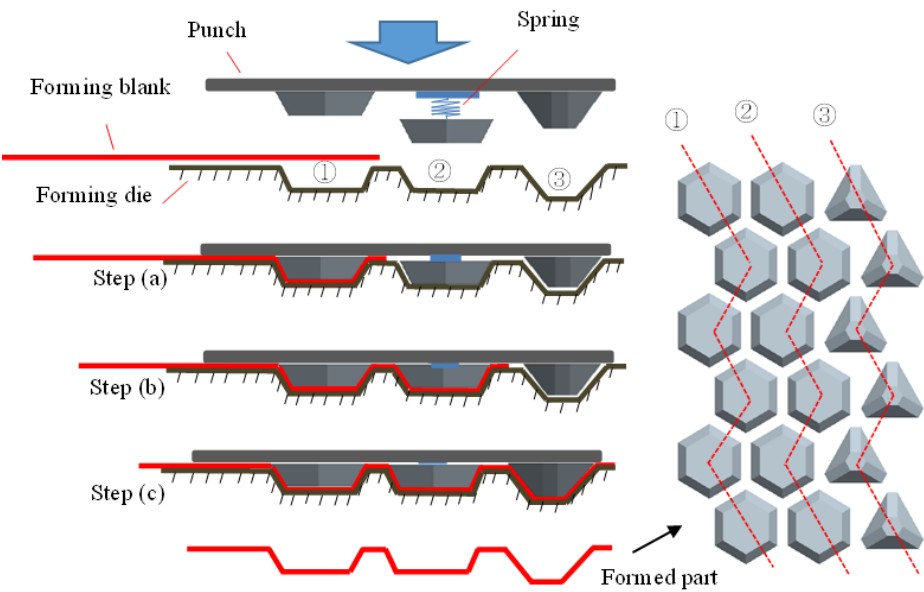

**Figure 6.** Forming procedure in progressive forming method.

As shown in step (c) of Figure 6, the formed plate is moved forward in one step, and the punch and die ② position the plate at position ② in the same way as in step (b). Simultaneously, the punch and die ① form the next intermediate model shape on the plate at position ①. At position ③, the intermediate model shape formed by the punch and die ③ can be formed into the shape of the truss core.

By sequentially repeating the above press-forming process, an entire truss core panel consisting of many truss cores can be fabricated.

The intermediate model shape in the progressive press-forming method was designed for easy forming. Although the shape of the intermediate model is not retained in the final product shape, and the design of the intermediate model is flexible, it has a significant impact on the molding performance of the molded product.

### 2.4. Intermediate Model Design Using Maximum Area Method

When preforming truss core panels using the progressive press method, it is advantageous to use as much material as possible to flow into the convexity of the intermediate model, thereby increasing the formable height of the final truss core. Herein, an intermediate model of a regular hexagonal pyramidal base as shown in Figure 7 was proposed.

The bottom shape of the truss core to be molded was arranged as indicated by the red line in Figure 7a. Centering on the molded truss core indicated by the shaded line, the maximum area in the regular hexagon indicated by the black line is available for the intermediate model in the vicinity of the truss core.

If the regular hexagon is reduced to approximately $0.5d$ as shown in Figure 7b, the base shape of the regular hexagon's intermediate model can be obtained. The length of its sides $a_i$ can be calculated using (1).

$$a_i = \frac{\sqrt{3}}{3}(c - d) \tag{1}$$

Here, the width of the adjacent gap of the intermediate model is determined by substituting the shape parameter $c$ of the truss core and the gap width $d = 8\,\text{mm}$, which

considers the strength conditions of the actual molding die, into Equation (1) to determine the side length $a_i = 16\,\text{mm}$ of the bottom shape of the intermediate model.

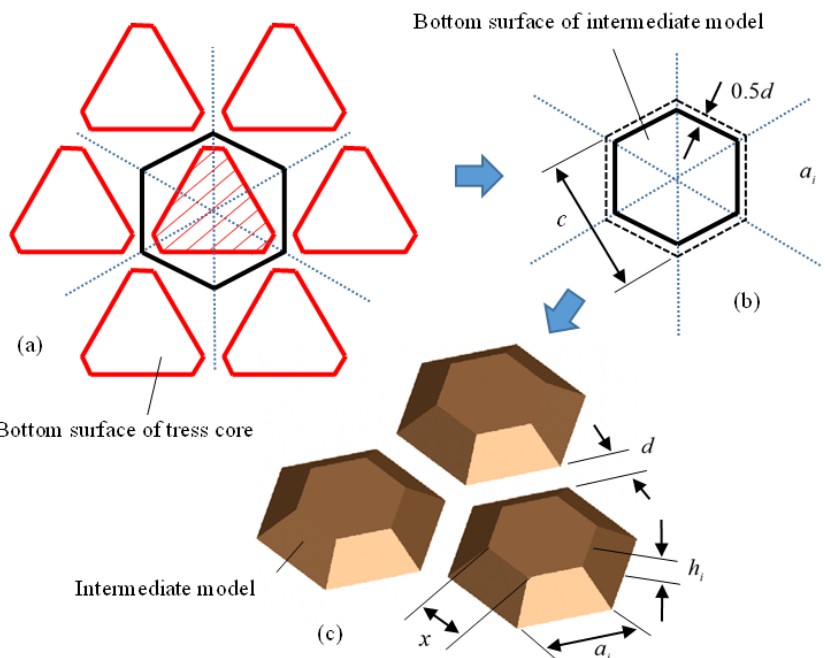

**Figure 7.** Intermediate model in progressive stamping method- the process of parametric design.

In addition, as shown in Figure 7c, the height $h_i$ of the intermediate model was set to half the height $h$ of the molded truss core (Figure 3), $h_i = 4.5\,\text{mm}$. The corner of the top face of the truss core panel was the most drastic area for thickness reduction. To reduce the thickness loss of this area as much as possible, the edge length b of the top face of the intermediate model can be determined by adjusting the surface area of the intermediate model to be equal to the surface area of the truss core.

In this section, using the geometric relationship of the three-dimensional shapes shown in Figures 5 and 7, the surface area $S_t$ of the truss core panel and the surface area $S_i$ of the intermediate model are expressed as

$$S_t = \frac{\sqrt{3}}{4}(a-b)\sqrt{(a-3b)^2 + 12h^2} + \sqrt{3}b\sqrt{(a-3b)^2 + 3h^2} + \frac{3\sqrt{3}}{2}b^2 \tag{2}$$

$$S_i = \frac{\sqrt{3}}{2}\left(c - d + \sqrt{3}x\right)\sqrt{\left(c - d - \sqrt{3}x\right)^2 + h^2} + \frac{3\sqrt{3}}{2}x \tag{3}$$

The shape parameters of the truss core are substituted into Equations (2) and (3), and when both equations are equal, the edge length $x = 10\,\text{mm}$ of the top surface of the intermediate model is obtained.

Therefore, the shape of the intermediate model shown in Figure 7c can be calculated by determining the side length $a_i$ of the bottom surface, side length $x$ of the top surface, and height $h$ and width $d$ of the adjacent gaps.

### 2.5. Verification of the Press-Forming Performance

To verify the forming performance of the intermediate model of the regular hexagonal pyramid proposed in the previous section when applied to the progressive press-forming method, the FEM (finite element method) analysis model created is shown in Figure 8. The analytical model comprises three parts: the punch (red), die (green), and formed plate material (light blue).

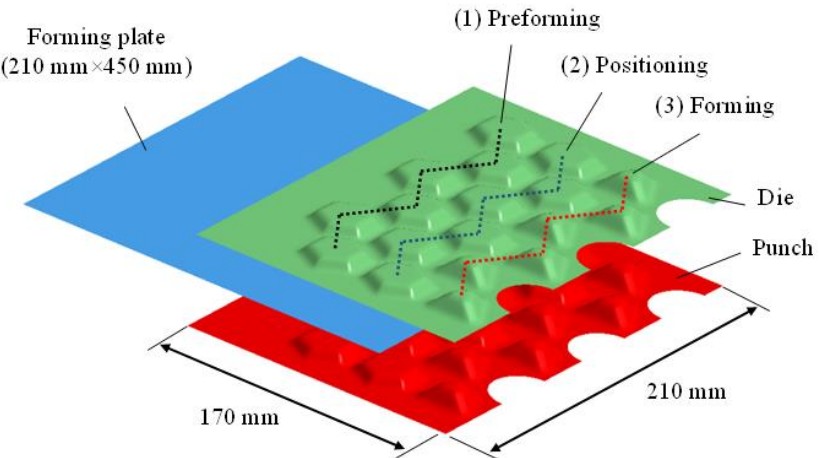

**Figure 8.** FEM model of progressive press forming by using hexagonal frustum intermediate model.

Three rows of punches and dies, including the pre-forming, positioning, and final forming rows, were set up along the feeding direction of the plate. The punches and dies were modeled using 4-node rigid shell elements with an average side length of 1.0 mm, and the formed plate was modeled using 4-node elastic–plastic shell elements with an average side length of 1.0 mm. The number of elements in the plate was 42,000 and the number of nodes was 42,411. The plate thickness is 1.0 mm, yield stress is 235 MPa, Young's modulus is 186.5 GPa, and Poisson's ratio is 0.3. The true stress and plastic strain diagram is shown in Figure 9. The forming load was applied by forced displacement.

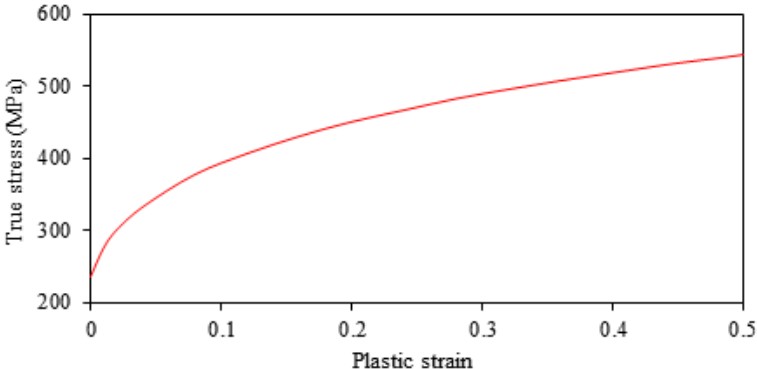

**Figure 9.** True stress–strain diagram of forming plate material.

Conversely, to verify the formability of truss core panels using the proposed intermediate model of regular hexagonal pyramids, the progressive press-forming die shown in Figure 10 was fabricated and actual forming experiments of truss core panels were conducted. Figure 10a shows a trial-forming test using a manual 30-ton hydraulic press (KPA-30, Osaka Jack, Japan) with a progressive press-forming die, and Figure 10b shows a trial-forming test using the progressive press-forming die. Figure 10b shows the forming die set fixed to the bolster of the press machine, and Figure 10c shows the punch set attached to the piston of the press machine.

In the forming die shown on the right side of Figure 10, three rows of forming die pairs are lined up from left to right, the first (black dotted line) and third (red dotted line) of which are for the preliminary intermediate model forming and final truss core forming, respectively. The second row (dotted blue line) is for positioning, and the mold block below is set higher than the other two rows by attaching a spring under it; thus, when the upper punch presses down, the positioning punch and mold of the second row first come into contact with each other. By clamping the intermediate core formed in the previous process, the plate is fixed firmly. As the upper punch pressed downward, the second row

of punches and die moved downward together. With the compression of the spring, the other two rows of punches and dies contacted and pressed the plate into a designed shape. At the end of each molding process, the spring under the second row of molds lifts the molded part back and separates the shaped plate from the mold.

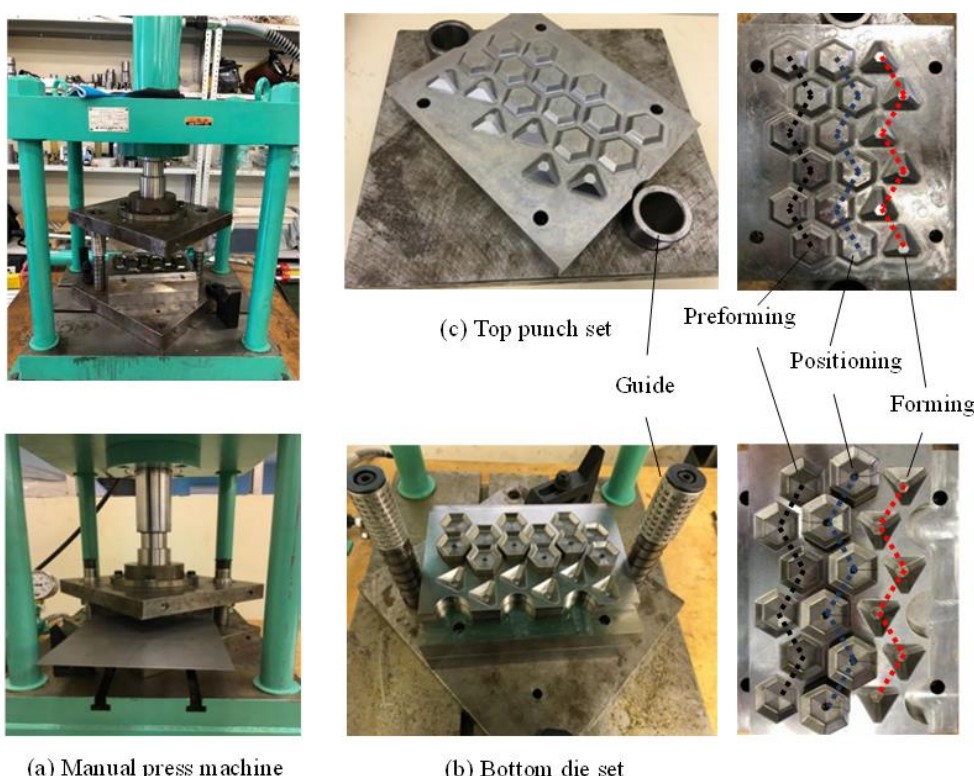

**Figure 10.** Experiment settings for truss score panel using the progressive press-forming method.

In addition, a mold set with cylindrical guides on both sides was utilized to maintain a constant and accurate positional relationship between the upper punch and lower forming die.

In progressive press forming using the proposed intermediate model of regular hexagonal pyramids, the thickness distribution of the formed truss core panels was used as an important index to evaluate the forming stability of the truss core panel. It is ideal to form a truss core panel with a uniform thickness distribution. To measure the thickness of the formed truss core panel, the panel was cut with a wire electrical discharge machine along the area to be measured, as shown in Figure 11b, and the thickness was measured using a dial caliper gauge (BO-1, Niigata Seiki of Japan, measurement accuracy 0.025 mm), as shown in Figure 11c. As shown in Figure 11a, it was possible to compare the thickness distribution of the truss core panel with the FEM analysis results and measured values of the thickness at the same site.

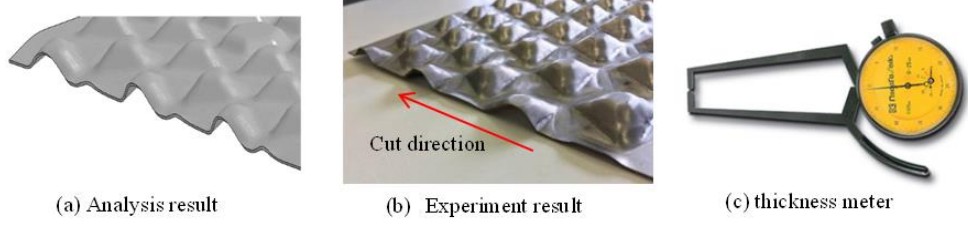

**Figure 11.** Method for measuring the plate thickness of the formed truss score panel.

## 3. Results and Discussion

### 3.1. Thickness Distribution of Formed Truss Core Panels

A contour diagram of the shape and thickness distribution of the truss core panel during progressive press forming is shown in Figure 12. In the first step, as shown in Figure 12a, an intermediate model of the first row is formed. Because the first forming area was close to the boundary of the plate and there was little forming resistance from the surrounding area, the maximum thickness reduction rate ($TR_{max}$) that occurred at the corner of the top face of the intermediate model was 17.6%.

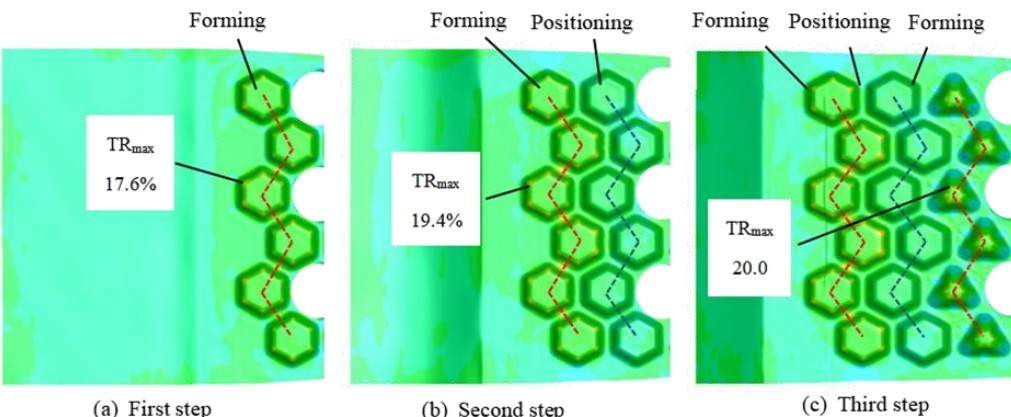

| (a) First step | (b) Second step | (c) Third step |

**Figure 12.** Truss core panel shape and thickness distribution during progressive press forming.

In the second step, shown in Figure 12b, the forming intermediate model shown by the blue dotted line is pressed first to accurately position the forming plate, and then the second row of intermediate models is formed. Compared with the first step, the maximum thickness reduction rate increased to 19.4% owing to the increased forming resistance from around the forming area.

In the third step, shown in Figure 12c, as in the second step, the formed intermediate model shown by the blue dotted line is first held in position, and then the truss core in the first row and the intermediate model in the third row are formed simultaneously. The maximum thickness reduction of 20.0% occurred at the top corner of the truss core.

A photograph of the truss core panel obtained via progressive press forming is shown in Figure 13. It can be observed that the truss core was formed into a good overall shape. In the flat part of the truss core panel, traces of the regular hexagon of the pre-formed intermediate model remained, implying that the truss core was performed on the entire plate surface, and this tendency is advantageous for improving the thickness reduction in the excessive thinning area of the truss core.

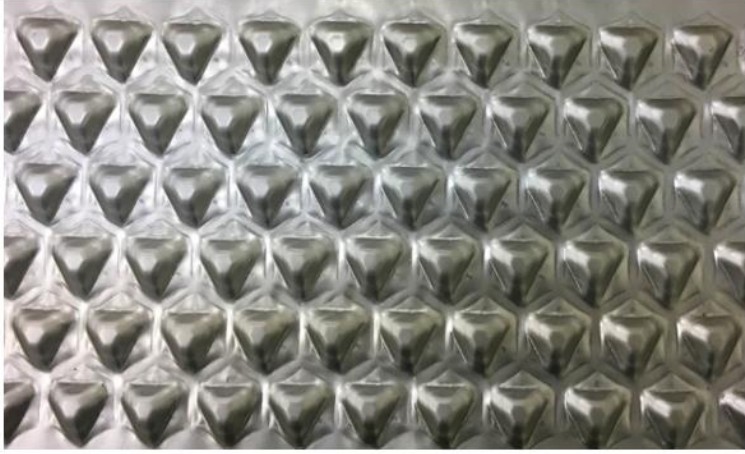

**Figure 13.** Formed truss score panel using the progressive press-forming method.

Figure 14 shows the evaluation results of the FLD (forming limit diagram) diagram obtained from progressive press-forming analysis. The figure shows that there are no points exceeding the forming limit (red), and all points are within the safe forming range (green), indicating that the proposed hexagonal pyramidal intermediate model can stably form truss core panels.

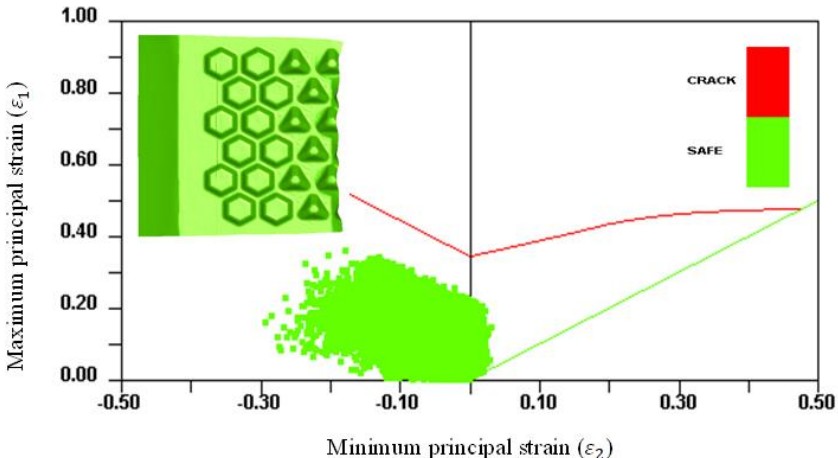

**Figure 14.** FLD evaluation result of truss score panel during the forming process.

### 3.2. Comparison between Analysis Results and Prototype Experiments

Figure 15 shows a comparison between the prototype experiment and the numerical analysis of the truss core panel obtained by progressive pressing. According to the figure, the maximum thickness reduction of the intermediate hexagonal pyramidal trapezoidal model obtained by pre-forming was 18.0% in the prototype experiment and 19.4% in the numerical analysis, whereas the maximum thickness reduction of the truss core obtained by final forming was 23.0% in the prototype experiment and 20.0% in the numerical analysis. The results of the prototype experiment and numerical analysis were in close agreement.

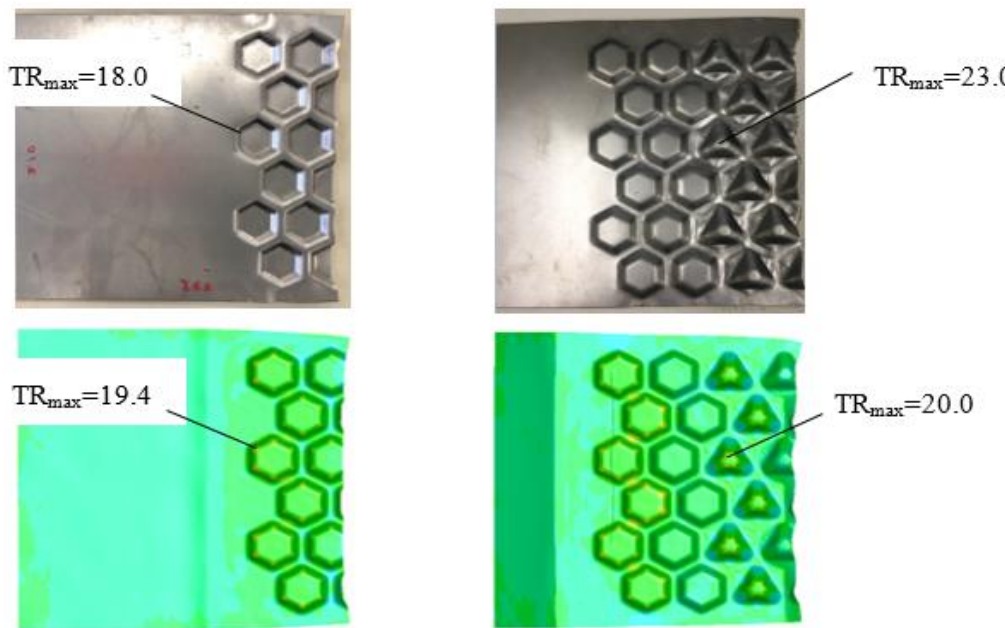

**Figure 15.** Plate thickness distribution of molded truss core panel.

For further discussion, only one truss core was removed from the molded truss core panel. The thickness measurement and analysis results for the partially molded shape are

summarized in Figure 16. The sampling points on the horizontal axis are the 15 points on the red dotted line along the truss core. The red dotted line shows the results of the prototype experiment and the blue dotted line shows the results of the numerical analysis.

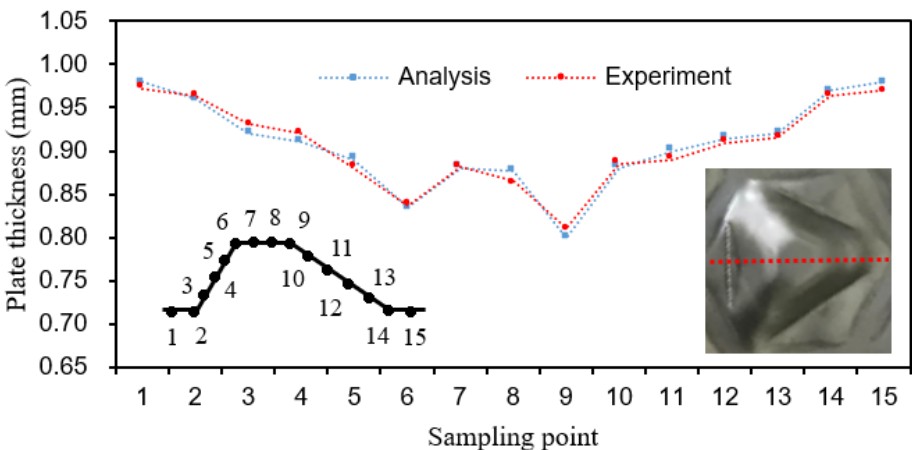

**Figure 16.** Plate thickness distribution of formed truss score panel.

From Figure 16, it can be observed that the thickness of the molded truss core decreased from the initial thickness of 1.0 mm, including the bottom points 1, 2, 14, and 15. It can be observed that the thicknesses of points 6 and 9, which are the tops of the truss cores, became the thinnest, and the thicknesses near the bottom points 3 and 13 became slightly thinner locally.

The volume of the plate before and after molding can be considered invariable. The surface area of the formed truss core panel will increase from the original plate; however, the thickness of the plate will decrease differently for different parts of the truss core panel. It is ideal to form a panel as uniformly as possible by applying an appropriate intermediate model in the progressive press-forming process. However, if there are any parts of the formed truss core panel where the thickness does not change before and after forming over the entire surface area of the panel, it is considered to be a bad pattern of press forming. Therefore, as shown in the results of Figure 16, there are no areas with the same thickness as the initial thickness of 1.0 mm over the entire surface of the truss core panel formed by the progressive press method, and all panels are thinner than the initial thickness, suggesting that the application of the intermediate model of the regular hexagonal pyramid proposed in this study is effective.

### 3.3. Comparison with Former Hemispherical Intermediate Model

To verify these results, a forming analysis of a truss core panel was performed using the hemispherical intermediate model studied in a paper [31] using the same plate material and forming conditions. The results of the thickness distribution analysis were compared with those of the forming analysis using the regular hexagonal pyramidal intermediate model proposed in this study.

Figure 17 shows an analytical model of the formation of a truss core panel using the progressive press method with a hemispherical intermediate model. The properties of the molded plate are the same as those of the analytical model shown in Figure 8, except that only the shape of the first row of the pre-formed plates and the positional relationship of the second row were changed.

Figure 18 shows a comparison of the results obtained by analyzing progressive press forming using the two different intermediate models. The blue dotted and red dotted lines show the results for the hexagonal pyramidal intermediate model and the hemispherical intermediate model, respectively.

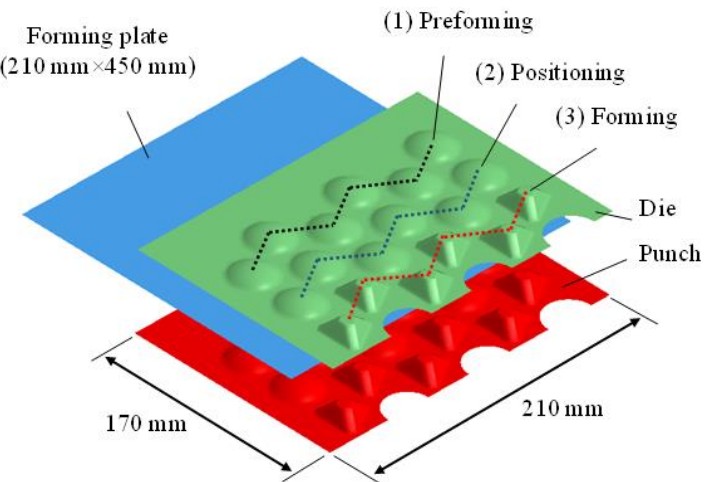

**Figure 17.** FEM model of progressive press forming using the hemispherical intermediate model.

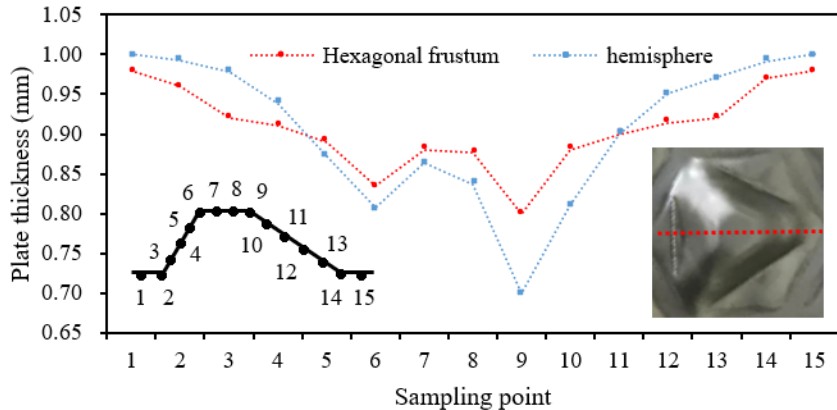

**Figure 18.** Plate thickness distributions of formed truss score panel.

Figure 18 compares the thickness distributions of the truss core panels formed using the two intermediate models and shows the following differences.

(1) Comparing the thickness at the bottom surface of the truss core shape (points 1, 2, 14, and 15), the thickness of the hemispherical intermediate model remains the same as the initial thickness of 1.0 mm, while the thickness of the bottom surface is reduced to 0.98 mm when formed with the hexagonal pyramidal intermediate model, indicating plastic deformation over the entire surface of the truss core panel.

(2) Along the sampling red dotted line of the truss core, the truss core formed by the hemispherical intermediate model was thinner near the center (points 5 to 11), and the truss core formed by the hexagonal pyramidal intermediate model was thinner in the left and right parts. The average thickness formed by the hemispherical intermediate model was 0.9073 mm, and the average thickness formed by the hexagonal pyramidal intermediate model was 0.9077 mm, which can be considered almost the same. The maximum thickness difference of truss core panel formed by the hexagonal pyramidal intermediate model is $1.00 - 0.70 = 0.30$ mm, while the maximum thickness difference of the truss core panel formed by the hexagonal pyramidal intermediate model is $0.98 - 0.80 = 0.18$ mm, indicating that the truss core panel formed by the hexagonal pyramidal intermediate model tends to have a more uniform thickness distribution.

(3) Comparing the thickness at point 9, the thinnest region of the formed truss core panels, the minimum thickness of the truss core panel formed by the hemispherical intermediate model is 0.70 mm and the minimum thickness of the truss core panel formed by the hexagonal pyramidal intermediate model is 0.80 mm, suggesting that cracking is more likely to happen when forming with the hemispherical intermediate model.

(4) The truss core panel formed with the hemispherical intermediate model has two locally thinned areas (points 6 and 9), whereas the preformed hexagonal pyramidal intermediate model has four locally thinned areas (points 3, 6, and 9) at the top edge. In contrast, the preformed hexagonal pyramidal intermediate model has a better forming property because the dissociation of weak areas in the molding process is expected to improve the molding stability.

## 4. Conclusions

In this study, a hexagonal pyramidal trapezoidal intermediate model for progressive press forming was proposed to improve the forming quality of truss core panels. The validity and practicality of the proposed hexagonal pyramidal trapezoidal intermediate model were verified through prototype experiments and numerical analysis.

During the design process of the intermediate model used in progressive press forming, the strength of the forming die was considered, and the area of the forming plate was designed to form the truss core as much as possible. It was confirmed that a decrease in the unreformed area of the forming plate before and after the pressing process can be an effective way to improve the forming quality of the truss core panel.

Compared to truss core panels formed by the conventional hemispherical intermediate model, the problem of the lack of plastic deformation in the local area of the formed plates is solved, and plastic deformation is observed over the entire surface of the truss core panel. This improves the uniformity of the overall thickness distribution. Compared with the truss core panel formed by the conventional hemispherical intermediate model, the minimum thickness of the truss core panel formed by the hexagonal pyramidal intermediate model is improved from 0.7 mm to 0.8 mm, confirming an improvement in the problem of excessive thinning in certain areas of the truss core panel.

**Author Contributions:** Writing—original draft, Z.T.; Writing—review & editing, Z.T. and X.Z.; Data curation, C.K.; Formal analysis, W.Z.; Investigation, J.G.; Software, C.K. and W.Z. All authors have read and agreed to the published version of the manuscript.

**Funding:** This research received no external funding.

**Institutional Review Board Statement:** Not applicable.

**Informed Consent Statement:** Not applicable.

**Data Availability Statement:** Not applicable.

**Conflicts of Interest:** The authors declare no conflict of interest.

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
