# Peer review of "Intermediate Model Design in the Progressive Stamping Process of a Truss Core Lightweight Panel"

_applsci, doi:10.3390/app12084002_

Round 1
Reviewer 1 Report
Review of the manuscript "Intermediate model design in progressive stamping process of truss core lightweight panel". A hexagonal pyramidal trapezoidal intermediate model for progressive press forming is proposed to improve the forming quality of truss core panels. Compared to truss core panels formed by the conventional hemispherical intermediate model, the problem of the lack of plastic deformation in the local area of the formed plates is solved, and plastic deformation is observed over the entire surface of the truss core panel. Also the uniformity of the overall thickness distribution is consequently improved.
The manuscript is interesting and well organized, english ok.
Without particular remarks the manuscript can be accepted after the following minor revisions:
Fig. 1 "Truss core" not "score".
Fig. 10 check a), b), c).
Fig. 11 check a), b), c).
Fig. 12 check a), b), c).
References (all) are not in compliance with the journal requirements.
Please increase readability of all figures.
Author Response
We appreciate your comments on the manuscript, and in response to your comments, we have made the corresponding changes, please see the attachment for details.

Reviewer 2 Report
This is an interesting article where the authors follow a current issue, namely the design of stamping process of truss core panel. Development of new materials by designing their inner architecture is an emerging trend in materials science and materials engineering [Yuri Estrin, et al., (2021) Architecturing materials at mesoscale: some current trends, Materials Research Letters, 9:10, 399-421, DOI: 10.1080/21663831.2021.1961908]. In reviewer view, design of new processes for stamping truss panel is a formidable challenge. The authors proposed a new stamp in the form of a hexagonal pyramidal trapezoidal intermediate model, and the efficiency of this proposal was proved both experimentally and numerically. Based on this, the reviewer recommends the article for publication.
The following minor issues should also be addressed before publication:
The authors should correct the text in the figures. For example, in Figure 10. the letter (a) occurs twice, and in Fig. 11 and Fig.12 - three times! Or, for example, the text in Figure 17 has moved out. Please check all figures and figure captions, as well as links to the corresponding figs. in the text.
Author Response

(The authors gave the same response as above.)

Reviewer 3 Report
- In the manuscript, is there any basis for the selection of truss core shape "a=35mm, b=5mm, c=35mm, h=9mm"? and the "h" font is wrong.
- The subject of "we" should be not used as much as possible.
- According to Fig. 7(b), please check the accuracy of Equation 1 carefully, if it is wrong, please correct it, and check other related calculation results.
- In the last line above Fig. 8, "3. Results" is redundant.
- There are errors in Figs. 10, 11 and 12, only (a), no (b) and (c).
- How to select the specific positions of the 15 points on the truss core, it should be explained through pictures.
- According to Figs. 16 and 18, it should be a blue dotted line, but lines 272 and 310 are written with a black dotted line.
Author Response

(The authors gave the same response as above.)

Reviewer 4 Report
In this study, a hexagonal pyramidal trapezoidal intermediate model for progressive press forming was proposed to improve the forming quality of truss core panels. The validity and practicality of the proposed hexagonal pyramidal trapezoidal intermediate model were verified through prototype experiments and numerical analysis. Compared to the conventional hemispherical intermediate model, the manufacturing process of the truss core panel using the proposed method was significantly improved. Overall, the paper structure is complete. The picture is clear and readable.
Author Response

(The authors gave the same response as above.)

Reviewer 5 Report
Dear Authors,
the paper is coherent and well written.
Some comments about the paper:
- Please explain acronyms when introduced for the first time, even the seemingly trivial ones.
- Please, check the numbering in the figures (in fig. 10, 11, and 12, there is only the index “(a)”!).
- There are some typos somewhere (e.g. line 173, "3. Results").
- Getting into the thickness measurement, are you sure your caliper could give the most appropriate measure, in particular in the corner zones?
- Figure 12 is not so clear. It could be useful to zoom in where you indicate the TRs.
- It could be useful, also, to insert a figure in which you indicate clearly the positioning of the 15 sampling points.
Author Response

(The authors gave the same response as above.)
